# Hallucination Mitigation in Natural Language Generation from Large-Scale Open-Domain Knowledge Graphs

**Xiao Shi, Zhengyuan Zhu, Zeyu Zhang, Chengkai Li**
Department of Computer Science and Engineering
The University of Texas at Arlington
Arlington, TX, USA
`{xiao.shi,zhengyuan.zhu,zeyu.zhang}@mavs.uta.edu, cli@uta.edu`

## Abstract

In generating natural language descriptions for knowledge graph triples, prior works used either small-scale, human-annotated datasets or datasets with limited variety of graph shapes, e.g., those having mostly star graphs. Graph-to-text models trained and evaluated on such datasets are largely not assessed for more realistic large-scale, open-domain settings. We introduce a new dataset, GraphNarrative, to fill this gap. Fine-tuning transformer-based pretrained language models has achieved state-of-the-art performance among graph-to-text models. However, this method suffers from information hallucination—the generated text may contain fabricated facts not present in input graphs. We propose a novel approach that, given a graph-sentence pair in GraphNarrative, trims the sentence to eliminate portions that are not present in the corresponding graph, by utilizing the sentence's dependency parse tree. Our experiment results verify this approach using models trained on GraphNarrative and existing datasets. The dataset, source code, and trained models are released at `https://github.com/idirlab/graphnarrator`.

## 1 Introduction

The task of *graph-to-text generation* aims to automatically produce natural language descriptions of knowledge graphs. A knowledge graph $\mathcal{G}$ stores factual information as subject-predicate-object triples, where each triple $(s, p, o)$ corresponds to an edge from the subject entity $s$ to the object entity $o$. The graph-to-text generation task entails, given a subgraph $G \subset \mathcal{G}$, generating a token sequence $(y_1, ..., y_n)$ to describe $G$. This task can be accomplished by constructing machine learning models (Clive et al., 2021; Castro Ferreira et al., 2019; Trisedya et al., 2018). The input to such a model is a graph itself—a small fragment of triples from a knowledge graph, as the outcome of some upstream operation, e.g., search, query and data

mining. The output is a textual sequence that describes the fragment of triples.

Verbalizing triples from knowledge graphs is crucial in a variety of tasks and applications, including systems created for querying knowledge graphs (Liang et al., 2021; Jayaram et al., 2016) as well as systems backed by knowledge graphs for question-answering (Zhou and Small, 2019; Ma et al., 2018) and fact discovery (Xian et al., 2019; Zhang et al., 2018). In these places, knowledge graph fragments must be conveyed to users in various forms, such as query results and discovered facts. Though a tiny part of a whole knowledge graph, such graph fragments can still be complex and thus challenging to comprehend. Instead, presenting them in natural language can help end users understand them better.

In graph-to-text generation, the preciseness and naturalness of the textual narration of graph fragments is important. Generating high-quality text can be particularly challenging for *large-scale* and *open-domain* knowledge graphs. Specifically, benchmark datasets in this line of research either are *hand-crafted* and *monotonous*, e.g., WebNLG (Gardent et al., 2017a) or only include *simple, special* formations in narrated input fragments, e.g., EventNarrative (Colas et al., 2021) and TEKGEN (Agarwal et al., 2021). Existing graph-to-text models, being trained and evaluated on these datasets, are largely not validated for more realistic large-scale, open-domain settings. Section 2 presents this analysis in detail.

This paper introduces GraphNarrative, a new dataset that fills the aforementioned gap between graph-to-text models and real-world needs. GraphNarrative consists of around 8.7 million (input graph, output text) pairs. The text in each pair is a Wikipedia sentence, whereas the corresponding graph comprises Freebase (Bollacker et al., 2008) entities and relationships described in the sentence. The large-scale of both Wikipedia and Freebase, the

linguistic variation in Wikipedia, and the complexity of sentences and corresponding graph structures make this dataset more aligned with real-world scenarios. For instance, GraphNarrative's 8.7 million input graphs are in 7,920 distinct topological shapes and 22% of the 8.7 million are star graphs, in contrast to 94% and 96% in EventNarrative and TEKGEN, respectively. Section 3 articulates the details of GraphNarrative's creation.

Given the demonstrated efficacy of fine-tuning pre-trained language models (PLMs) in producing state-of-the-art results on graph-to-text (more details in Section 4), we adopt the same approach. As pointed out in (Agarwal et al., 2021; Dušek et al., 2018), though, this approach may suffer from information *hallucination*, i.e., the output texts may contain fabricated facts not present in input graphs. For example, given a two-triple input graph {(Neff Maiava, *date of birth*, 01 May 1924), (Neff Maiava, *date of death*, 21 April 2018)}, (Agarwal et al., 2021) reported their model generates "Neff Maiava (1 May 1924 - 21 April 2018) was an Albanian actor." Not only the input does not mention Maiava's profession or citizenship, but also in the real-world he was an American Samoan wrestler instead.

Very few have considered how to mitigate hallucination in graph-to-text generation, except for (Agarwal et al., 2021; Wang et al., 2021; Ma et al., 2022). The first two studies attempted to address hallucination by further fine-tuning PLMs on WebNLG after fine-tuning on noisier automatically-extracted datasets. (Ma et al., 2022) adopted a different approach, by filtering out training instances when the ROUGE-1 (Lin, 2004) scores between the input and the output fall below a certain threshold. However, these studies did not quantify the prevalence of hallucination in their models' outputs. Nor did they provide direct experiment results or other evidence to verify the approach in reducing hallucination. We are the first to quantitatively measure the prevalence of hallucination in graph-to-text. We also developed a novel approach to mitigating hallucination by aiming at the problem's root—mismatch between graph and text in training data. Given a graph-text pair in GraphNarrative, the approach trims the text, i.e., a Wikipedia sentence, by eliminating portions not represented in the graph. This process, named *sentence trimming*, is accomplished by analyzing the shortest paths between graph entities within the sentence's dependency parse tree (details in Section 5).

We conducted comprehensive automatic and human assessments of text descriptions generated by fine-tuned PLMs, specifically BART (Lewis et al., 2020) and T5 (Raffel et al., 2020). The automatic evaluation results consistently demonstrated that models performed better with the use of sentence trimming, across the datasets of GraphNarrative, TEKGEN, WebNLG, and DART (Nan et al., 2021). The approach led to the increment of 12 and 7 points in BLEU score (Papineni et al., 2002) for GraphNarrative and TEKGEN, respectively. A T5-large model fine-tuned on GraphNarrative with sentence trimming achieved state-of-the-art results on the WebNLG benchmark. Furthermore, human evaluation results showed that sentence trimming on average reduced 1.4 entity hallucinations and 1 relationship hallucination per text description.

The contributions of this paper are as follows.

• A new dataset, GraphNarrative, that fills the gap between existing datasets and large-scale real-world settings.

• The first to quantify hallucinations produced by graph-to-text models.

• A novel approach, sentence trimming, to hallucination mitigation.

• Comprehensive experiments and evaluations that verify the quality and utility of GraphNarrative, as well as the effectiveness of sentence trimming.

## 2 Limitations of Existing Datasets

*First*, most previous models were trained on small hand-crafted datasets that contain limited entity types and relations. For instance, WebNLG includes 2,730 distinct entities and 354 distinct relations. In contrast, real-world knowledge graphs can be much larger. For example, according to (Heist et al., 2020), Wikidata (Vrandečić and Krötzsch, 2014) has 52,252,549 entities, 2,356,259 classes, 6,236 relations, and 732,420,508 triples. The hand-crafted approach cannot scale to these massive knowledge graph, as it is impossible to manually write training graph-text pairs for so many different entity types, relations, and topic domains.

*Second*, the text descriptions in hand-crafted datasets such as WebNLG tend to follow monotonous templates, plausibly because the examples were written by a small number of human contributors. This limits the capability of trained models to use diverse expressions in narrating graph fragments. This lack of linguistic variation can hamper the usability of a text generation system.

*Third*, the graph fragments in existing datasets are largely limited to simple *star graphs* (each graph consisting of a center entity and some of its one-hop neighbors) or more general acyclic graphs (i.e., one or more trees). The graphs in WebNLG have 41 distinct topological shapes (Appendix D), out of which 32 are acyclic graphs. The cycles are all 2-edge loops or self-loops. In DART, 83% of the graphs are star graphs. In automatically-generated datasets EventNarrative and TEKGEN, 94% and 96% of the graphs are star graphs, respectively. Another automatically-collected dataset, AGENDA (Koncel-Kedziorski et al., 2019), has only 2% star graphs. But it only contains 7 distinct relations in the special domain of scientific research. On the contrary, in practical scenarios the input fragments can be of *complex, general* rather than simple, special formations. While direct measurement is lacking, we used the graphs described in Wikipedia sentences as a proxy for gauging the shape diversity of graphs that need to be narrated. We manually analyzed the formations of graphs presented in 100 random Wikipedia sentences, and we found only 39 of the 100 graphs are star graphs. Similar but automatic analysis of the complete Wikipedia corpus (more details in Section 3, Figure 2) show that only 2 of the 10 most frequent graph formations [1] are star graphs, and 3 are cyclic graphs.

## 3 The GraphNarrative Dataset

This section explains how we generated our new dataset GraphNarrative by aligning Wikipedia texts with Freebase. Note that the methodology could be applicable to text corpora beyond Wikipedia and knowledge graphs beyond Freebase. This section also contrasts GraphNarrative with existing benchmark datasets to demonstrate how it addresses current datasets' limitations.

### 3.1 Dataset Creation: Graph-Text Alignment

For each applicable Wikipedia sentence $W$, we create the corresponding subgraph $G$ in Freebase, to form a graph-sentence pair $(G, W)$ as one example instance in the dataset. See Figure 1 for an example. This is achieved by an *entity linking* step followed by an *edge detection* step.

*Entity linking*. It maps a span of tokens in the Wikipedia sentence $W$ to an entity $e$ in Freebase.

---

[1]Or 3 out of the 10, depending on whether considering a 3-node path as a star or not.

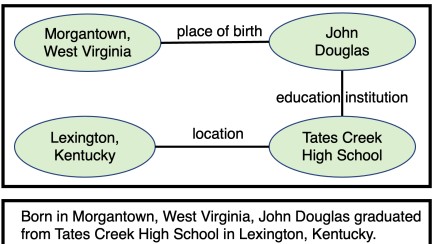

Figure 1: A graph-sentence pair in GraphNarrative

Our customized entity linking solution consists of coreference resolution (McCarthy and Lehnert, 1995), wikification (Csomai and Mihalcea, 2008), and Wikipedia-to-Freebase entity mapping. The entity mapping (more details in Section B.3) created 4,408,115 one-to-one mappings between English Wikipedia entities (i.e., articles) and Freebase entities, through a combination of three engineering methods—by using existing mapping in Freebase, by using Wikidata as the midpoint connecting Wikipedia and Freebase entities, and similarly by using DBpedia (Auer et al., 2007) as the midpoint. For *wikification*, our simple approach maps a span of tokens in a Wikipedia article $D$ to a Wikipedia entity, if the tokens exactly match either the entity's full title or any of the entity's wikilink anchor text in the same article $D$. For *coreference resolution*, we applied the implementation (Lee et al., 2017) in AllenNLP (Gardner et al., 2017) on Wikipedia articles to replace pronouns and aliases with corresponding entities. The results of aforementioned processes were put together—a Wikipedia entity appearance in a Wikipedia sentence, either originally as a wikilink or detected through wikification upon coreference resolution, leads to the detection of the corresponding Freebase entity via the mapping results.

*Edge detection*. Given the Freebase entities detected from a Wikipedia sentence $W$, it identifies Freebase edges between the entities such that the corresponding relations are described in $W$. Given a pair of such entities, if Freebase contains only one edge between them, our simple method assumes the corresponding relationship is described in $W$. If Freebase has multiple edges between them, we include the edge whose label tokens overlap with $W$. If there are still multiple such edges, we include the edge that is most frequent in Freebase. All these detected edges form the graph $G$ that pairs with $W$ as an instance $(G, W)$ in the dataset. Note that the simple assumptions in this approach may lead to both false positives and false negatives. In practice, the resulting dataset has solid quality

| Dataset | Knowlege Graph | Text Corpus | Domain | Instances | Entities | Triples | Relation | Star Graphs |
|---|---|---|---|---|---|---|---|---|
| WebNLG | DBpedia | Handcraft | 15 DBpedia categories | 25,298 | 2,730 | 3,221 | 354 | 57% |
| DART | N/A | Handcraft | N/A | 38,391 | 27,000 | 32,139 | **3,834** | 83% |
| AGENDA | N/A | Scientific abstract | Scientific research | 40,720 | 159,691 | 177,568 | 7 | **2%** |
| EventNarrative | Wikidata | Wikipedia | Events | 224,428 | 305,685 | 649,337 | 672 | 94% |
| TEKGEN | Wikidata | Wikipedia | Open domain | 7,895,789 | **4,856,439** | 11,373,838 | 663 | 96% |
| GraphNarrative | Freebase | Wikipedia | Open domain | **8,769,634** | 1,853,752 | **15,472,249** | 1,724 | 22% |

Table 1: Comparison of graph-to-text datasets

(detailed assessment in Section 6.2). Nevertheless, our workflow of dataset creation allows for more advanced and accurate methods in each component.

## 3.2 Characteristics of GraphNarrative

This section qualitatively and quantitatively analyzes how GraphNarrative bridges the gap between graph-to-text models and real-world settings.

*Scale and variety of entities and relations.* GraphNarrative contains 8,769,634 graph-sentence pairs, 1,853,752 entities, 15,472,249 triples, and 1,724 relations from 84 Freebase domains (see Appendix B.1). As Table 1 shows, most other datasets are significantly smaller in these aspects.

*Linguistic variation.* Using Wikipedia as the corpus, the graph-text pairs in GraphNarrative allow a model to learn from many Wikipedia authors' diverse narrations. On the contrary, text in handcrafted datasets such as WebNLG and DART tend to follow monotonous templates from a small number of human contributors.

*Graph structure complexity.* The graphs in GraphNarrative contain 1–15 triples and 2–20 entities, in 7,920 distinct topological shapes based on graph isomorphism. (Detailed distributions of graph instances and shapes are in Appendix B.2.) Figure 2 displays the 10 most frequent shapes along with their instance counts. Furthermore, only 22% of the instance graphs are star graphs. On the contrary, EventNarrative and TEKGEN are dominated by star graphs, as Table 1 shows.

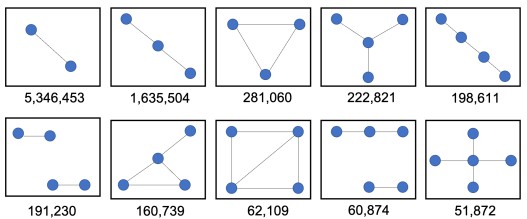

Figure 2: 10 most frequent graph shapes in GraphNarrative, with instance counts

## 4 Models

Existing graph-to-text models often use a decoder-only (Brown et al., 2020) or encoder-decoder structure (Sutskever et al., 2014), where the encoder

learns representations of input graphs and the decoder subsequently translates the representations into token sequences. In decoder-only models, e.g., GPT-2 (Radford et al., 2019), GPT-3 (Brown et al., 2020) and ChatGPT (i.e., GPT-3.5) (OpenAI, 2022) in the GPT family, the decoder uses text sequence embedding as the representation of input graphs. Encoder-decoder models fall into two main categories based on graph representations— 1) sequence-to-sequence models that encode linearized graphs' token sequences with LSTMs (Trisedya et al., 2018; Gardent et al., 2017b) or Transformers (Castro Ferreira et al., 2019), and 2) models that use a dedicated graph encoder to capture the structural information of knowledge graphs (Schmitt et al., 2021; Ribeiro et al., 2020; Marcheggiani and Perez-Beltrachini, 2018).

Many of the aforementioned models fine-tune PLMs. Fine-tuning transformer-based, sequence-to-sequence encoder-decoder PLMs (e.g., T5 and BART) achieved state-of-the-art performance on WebNLG (Ribeiro et al., 2021; Wang et al., 2021; Clive et al., 2021) and DART (Aghajanyan et al., 2021). (Yuan and Färber, 2023) reported that fine-tuning T5 and BART on WebNLG and AGENDA datasets yielded better results than zero-shot learning using GPT-3 and GPT-3.5. They also reported factual hallucinations from GPT-3 and GPT-3.5. While the reported results from fine-tuning GPT-2 on WebNLG (Harkous et al., 2020) are worse than current state-of-the-art, no results have been reported based on fine-tuning GPT-3 or GPT-3.5.

Following the state-of-the-art approach, we also fine-tuned T5 and BART on GraphNarrative and other datasets in comparison. In training and applying a graph-to-text model, an instance graph is linearized into a token sequence. Following the method in (Ribeiro et al., 2021), the graph in Figure 1 would be linearized as "<H> John Douglas <R> place of birth <T> Morgantown, West Virginia <H> John Douglas <R> education institution <T> Tates Creek High School <H> Tates Creek High School <R> location <T> Lexington, Kentucky" where the special tokens <H>, <R> and <T> denote subjects, relations and objects, respectively.

## 5 Mitigation of Hallucination

The culprit of the hallucination problem discussed in Section 1 is fabrication in training data—textual descriptions containing information not found in input graphs. This is evidenced by that, while graph-to-text models frequently produce hallucination when trained on TEKGEN, it rarely happens on WebNLG. Hallucinated facts are seldom found in the clean, manually-crafted WebNLG but are present in automatically extracted graph-text pairs in TEKGEN due to extraction errors.

There could be two plausible directions in tackling graph-to-text hallucination. One is to improve our graph-text alignment method (Section 3.1). The graph extracted from a piece of text during alignment may miss certain entities or relationships due to either extraction errors or disparities between the text corpus and the knowledge graph. The resulting graph-text pair may misguide the trained model to hallucinate facts. A more accurate alignment method can reduce such erroneous pairs and thereby reduce hallucination. However, this method has an inherent limitation—since a knowledge graph in real-world is often far from complete, there will be facts in text that cannot be mapped to the knowledge graph. Nevertheless, in principle, a way to combine this approach with the other approach discussed below is open for investigation.

This study explores a different direction in mitigating hallucination. Given a (Freebase subgraph $G$, Wikipedia sentence $W$) pair produced by alignment, we introduce a *sentence trimming* algorithm (Algorithm 1 in Appendix A) to turn $W$ into a trimmed sentence $W_{trim}$ by eliminating portions that are not present in $G$ while preserving the sentence's main idea. Below we provide a sketch of the algorithm, while keeping its pseudo code and description in Appendix A.

First, the algorithm parses $W$ and generates its dependency parse tree (DPT) $W_{tree}$, using spaCy (Honnibal et al., 2020). Then, for each triple $t_i = (s_i, p_i, o_i) \in G$, it identifies the shortest dependency path (SDP) between $s_i$ and $o_i$, i.e., the shortest path between the two entities' tokens in $W_{tree}$. It then finds the leftmost position index $min\_pos$ in sentence $W$ among all tokens on all triples' SDPs, and similarly the rightmost position index $max\_pos$. This process results in the trimmed sentence $W_{trim}$, a sub-sequence of $W$ spanning from $min\_pos$ to $max\_pos$.

An example is in Figure 3 which illustrates the

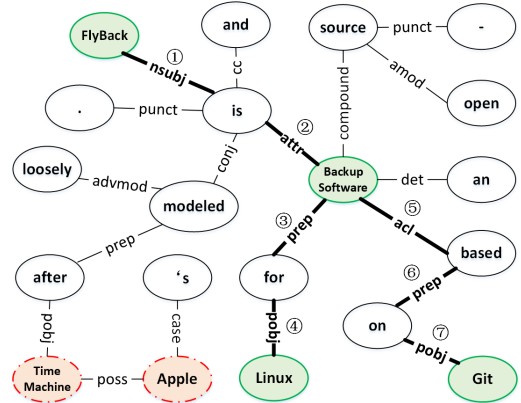

Figure 3: Dependency parse tree of sentence "FlyBack is an open-source Backup Software for Linux based on Git and modeled loosely after Apple's Time Machine."

DPT of the sentence $W$ in its caption. The corresponding graph $G$ from the graph-text alignment process is {(FlyBack, *software_genre*, Backup Software), (FlyBack, *operating_system*, Linux), (FlyBack, *basis*, Git)}. Note that entities Apple and Time Machine in $W$ are missing from $G$. The SDPs for the three triples are (①, ②), (①, ②, ③, ④), and (①, ②, ⑤, ⑥, ⑦), respectively. Given the SDPs, $min\_pos$ is attained by FlyBack and $max\_pos$ is attained by Git. Hence, $W_{trim}$ is "FlyBack is an open-source Backup Software for Linux based on Git". The sequence "and modeled loosely after Apple's Time Machine.", related to the missing entities Apple and Time Machine, is trimmed from $W$.

Note that, a regular DPT will break up entities such as Backup Software into individual tokens, each for a node in the DPT. To avoid that, we used a modified concept of DPT—we preprocessed entity names and tokenized each entity's name into a single token. Specifically, the two tokens Backup and Software were combined into token BackupSoftware.

## 6 Results

### 6.1 Datasets

We performed experiments on four datasets: GraphNarrative, TEKGEN (the large-scale, open-domain graph-to-text dataset that resembles ours the most), and WebNLG and DART (two human-annotated datasets). Detailed statistics about these and other datasets can be found in Table 1.

• GraphNarrative is partitioned into training, development and test sets in accordance with the process elaborated below. Each edge in Freebase belongs to a topic domain. Every instance in GraphNarrative, i.e., a (graph, sentence) pair, is assigned a domain, using the most frequent domain among the graph's edges. We then divided

GraphNarrative into *seen* and *unseen* partitions according to the numbers of instance pairs in different domains. Domains with very few (less than 2,000) pairs were designated as unseen domains, while the remaining domains are seen. A full list of the seen and unseen domains is in Appendix B.1. All instances in unseen domains go to test set. In the seen partition, 90%, 5% and 5% of the instances are allocated for training, development and test, respectively. This resulted in 7,880,214 instances in the training set, 437,514 in the development set, and 451,906 in the test set, including 13,453 instances from unseen domains. Having unseen instances in the test set helps us evaluate models' generalization ability. Choosing domains with limited discussion ensures that the model has encountered only a few such instances during pre-training of PLMs.

• In TEKGEN, each instance pair contains a Wikipedia sentence and a Wikidata subgraph extracted from the sentence. We used the original training, development and test set partitions from (Agarwal et al., 2021). We could not use all instances due to lack of mappings between entity names and their surface texts. Without such information sentence trimming cannot be applied. To maximize the utility of available instances, we used aliases sourced from TEKGEN and leveraged regular expressions to identify time and people's names. Consequently, we obtained 3,811,288 instances for training, 476,439 for development, and 484,958 for test, out of the original 6,310,061, 788,746, and 796,982 instances, respectively.

• In the standard WebNLG 2017 challenge dataset, each instance is composed of a graph from DBpedia and one or multiple sentences written by human annotations to describe the graph's content. Its test set is divided into the *seen* partition, which contains 10 DBpedia categories present in the training and development sets, and the *unseen* partition, which covers 5 categories absent from the training and development sets. We used the same partitioning as in the dataset.

• DART is a data-to-text dataset that comprises pairs of (triple-set, sentence) gathered from a variety of sources, including WebNLG, E2E (Novikova et al., 2017), and sentences collected through crowdsourcing and paired with tables extracted from WikiSQL (Zhong et al., 2017) and WikiTableQuestions (Pasupat and Liang, 2015). We used the original partitioning of training, development and test sets in DART.

| ST | Hallucinated Entities | Missed Entities | Hallucinated Relations | Missed Relations | Grammar |
|---|---|---|---|---|---|
| w/o | 1.163 | 0.003 | 1.340 | 0.040 | 4.793 |
| w/ | 0.306 | 0.003 | 0.453 | 0.083 | 4.613 |

Table 2: Human evaluation of GraphNarrative quality

## 6.2 Human & Automatic Evaluation Metrics

*Human evaluation metrics.* We evaluated the quality of both the GraphNarrative dataset and the sentences generated by models, focusing on whether sentences in the dataset or produced by models fabricate facts that are not in the corresponding graphs narrated by the sentences. To the best of our knowledge, no prior study has quantitatively evaluated the quality of graph-to-text datasets or models with regard to hallucination. Specifically, we define the following four metrics: numbers of *hallucinated entities* (entities not present in the graph but mentioned in the sentence), *missed entities* (entities present in the graph but not mentioned in the sentence), *hallucinated relations* (relations not present in the graph but mentioned in the sentence), and *missed relations* (relations present in the graph but not mentioned in the sentence).

In addition, we also evaluated the quality of sentences using average *grammar errors* per sentence, on a scale of 1-5: 5 (no errors), 4 (one error), 3 (two to three errors), 2 (four to five errors), and 1 (more than five errors).

*Automatic evaluation metrics.* For model-generated sentences, we also report automatic evaluation results using standard natural language generation metrics BLEU (Papineni et al., 2002), METEOR (Banerjee and Lavie, 2005) and chrF++ (Popović, 2015).

## 6.3 Experiment and Evaluation Results

**1) GraphNarrative dataset quality.** Three human annotators evaluated the quality of the graph-sentence pairs in GraphNarrative. We randomly chose 100 pairs, where each sentence has the original version and the trimmed version using the algorithm in Section 5. The total 200 pairs were then shuffled so that annotators cannot tell whether a sentence is original or not. Each human annotator scored all 200 pairs using the metrics in Section 6.2, and their scores were averaged.

Table 2 presents the results. In this and subsequent tables, sentence trimming is denoted ST. The average hallucinated entities and relations [2] per

---

[2] We used the metrics defined in Section 6.2 for evaluating both dataset quality and model output. This may cause confusions in understanding the measures in the context of

| Model | ST | BLEU | | | METEOR | | | chrF++ | | |
|---|---|---|---|---|---|---|---|---|---|---|
| | | all | seen | unseen | all | seen | unseen | all | seen | unseen |
| BART-base | w/o | 33.18 | 33.33 | 27.52 | 17.18 | 17.26 | 14.63 | 36.56 | 36.74 | 30.75 |
| BART-base | w/ | 46.49 | 46.77 | 36.67 | 24.43 | 24.53 | 21.30 | 49.92 | 50.12 | 43.29 |
| BART-large | w/o | 32.35 | 32.48 | 27.56 | 17.45 | 17.53 | 15.07 | 37.12 | 37.29 | 31.58 |
| BART-large | w/ | 46.04 | 46.18 | 40.98 | 24.35 | 24.41 | 22.17 | 49.69 | 49.85 | 44.72 |
| T5-small | w/o | 19.48 | 19.53 | 17.34 | 15.78 | 15.85 | 13.79 | 33.92 | 34.08 | 28.90 |
| T5-small | w/ | 43.72 | 43.87 | 38.11 | 23.40 | 23.48 | 21.10 | 48.15 | 48.31 | 42.65 |
| T5-base | w/o | 16.89 | 16.95 | 14.63 | 16.23 | 16.30 | 14.10 | 35.37 | 35.54 | 29.84 |
| T5-base | w/ | 42.18 | 42.29 | 37.85 | 24.20 | 24.27 | 21.94 | 49.63 | 49.80 | 44.18 |
| T5-large | w/o | 22.22 | 22.26 | 20.41 | 17.16 | 17.23 | 15.02 | 36.78 | 36.95 | 31.40 |
| T5-large | w/ | 45.12 | 45.16 | **43.40** | **24.77** | 24.84 | **22.54** | **50.44** | 50.60 | **45.21** |

Table 3: Model performance on GraphNarrative

| Model | WebNLG | | | DART | | |
|---|---|---|---|---|---|---|
| | BLEU | METEOR | chrF++ | BLEU | METEOR | chrF++ |
| T5-large | 4.01 | 9.54 | 24.64 | 3.44 | 7.93 | 23.17 |
| Filter-T5 | 19.81 | 30.36 | 54.01 | 16.15 | 27.53 | 48.69 |
| GN-T5 | 21.38 | 31.82 | 56.83 | 19.35 | 27.35 | 50.41 |
| GNST-T5 | 27.60 | 32.27 | 56.81 | 19.42 | 28.07 | 50.96 |

Table 4: Zero-shot performance of models on WebNLG and DART test sets

graph-sentence pair are 1.163 and 1.340, respectively. This reflects the challenges in graph-to-text alignment and the source of hallucination, as explained in Section 5. Applying sentence trimming reduced these numbers to 0.306 entities and 0.453 relations, clearly showing its effectiveness in enhancing graph-text alignment. On the other hand, when graphs were extracted from corresponding sentences to form GraphNarrative, information not present in the sentences was seldom introduced into the graphs, as reflected in the small missed entities and relations, both less than 0.1. Sentence trimming only slightly increased missed relations from 0.040 to 0.083, showing insignificant side effect of removing from sentences information covered in corresponding extracted graphs. With regard to grammar, while sentence trimming led to a slight decline in the grammar score, the difference (4.793 vs. 4.613) is not substantial.

**2) Model performance on** GraphNarrative. We fine-tuned various T5 (small: 60M parameters, base: 220M parameters, and large: 770M parameters) and BART (base: 140M parameters, large: 400M parameters) models on GraphNarrative for $10^6$ steps with a batch size of 8 using the Adam optimizer (Kingma and Ba, 2014) and an initial learning rate of $3 \times 10^{-5}$. We employed a linearly decreasing learning rate schedule without warm-up and set the maximum target text length to 384 tokens. Our implementation was based on (Ribeiro

et al., 2021), which adapted PLMs from Hugging Face (Wolf et al., 2019) for graph-to-text. The automatic evaluation results of different models on GraphNarrative are in Table 3. Fine-tuning the T5-large model attained the best performance across most metrics, consistent with findings on WebNLG in (Ribeiro et al., 2021; Wang et al., 2021).

**3)** GraphNarrative **in enhancing generalization ability**. To assess if GraphNarrative may enhance PLMs' generalization ability, we conducted both zero-shot learning and fine-tuning experiments employing GN-T5 and GNST-T5 on WebNLG and DART, where GNST-T5 denotes the fine-tuned T5-large model on GraphNarrative with sentence trimming, and GN-T5 denotes the counterpart without sentence trimming. They are also compared with the original T5-large model as a point of reference.

*Zero-shot results.* For zero-shot learning, we directly applied the above-mentioned three models on the test sets of WebNLG and DART. The results are in Table 4. The results reveal that fine-tuning PLM on GraphNarrative substantially improves its generalization capabilities.

*Fine-tuning results.* We subjected the three models to further fine-tuning on WebNLG and DART for 100 epochs with an early stopping patience of 20 epochs, while keeping other hyperparameters consistent with those in Part 2, Section 6.3. No trimming was performed on WebNLG and DART, as their sentences were authored by human annotators, with very few hallucinated or missed entities and relations. Table 5 compares the performance of different graph-to-text models on WebNLG test set, including the reprint of the results from seven prior studies. GNST-T5 fine-tuned on WebNLG outperformed others on most metrics, particularly in the unseen category. This improvement suggests that GraphNarrative enhances the generalization ability of PLMs. Table 7 shows the fine-tuning results on DART test set. The model performance improvement by sentence trimming is not obvious. This is further discussed in Part 4, Section 6.3.

---

dataset quality—a *hallucinated* entity refers to an entity from the original sentence that is *missed* in the corresponding extracted graph! We decided to tolerate this potential confusion for the sake of consistent metric definition.

| Model | BLEU | | | METEOR | | | chrF++ | | |
|---|---|---|---|---|---|---|---|---|---|
| | all | seen | unseen | all | seen | unseen | all | seen | unseen |
| (Gardent et al., 2017b) | 33.24 | 52.39 | 6.13 | 23.00 | 37.00 | 7.00 | - | - | - |
| (Marcheggiani and Perez-Beltrachini, 2018) | 55.90 | - | - | 39.00 | - | - | - | - | - |
| (Ferreira et al., 2019) | 51.68 | 56.35 | 38.92 | 32.00 | 41.00 | 21.00 | - | - | - |
| (Ribeiro et al., 2020) | - | 63.69 | - | - | 44.47 | - | - | 76.66 | - |
| (Ribeiro et al., 2021) | 59.70 | 64.71 | 53.67 | 44.18 | 45.85 | **42.26** | 75.40 | 78.29 | 72.25 |
| (Wang et al., 2021) | 60.56 | 66.07 | 53.87 | 44.00 | 46.00 | 42.00 | - | - | - |
| (Aghajanyan et al., 2021) | 56.30 | 64.80 | 46.10 | 42.00 | 46.00 | 38.00 | - | - | - |
| GNST-T5 (ours) | **61.46** | **66.49** | **55.35** | **44.30** | **46.23** | 42.08 | **76.20** | **79.35** | **72.76** |

Table 5: Performance comparison of different graph-to-text models on WebNLG test set

| Dataset | ST | BLEU | | | METEOR | | | chrF++ | | |
|---|---|---|---|---|---|---|---|---|---|---|
| | | all | seen | unseen | all | seen | unseen | all | seen | unseen |
| TEK | w/o | 60.43 | 65.49 | 54.32 | 44.06 | 46.04 | 41.90 | 75.73 | 78.83 | 70.13 |
| GEN | w/ | 60.82 | 65.42 | 55.12 | 44.25 | 46.13 | 42.18 | 76.16 | 79.11 | 72.35 |
| Graph | w/o (GN-T5) | 60.26 | 65.44 | 54.06 | 44.08 | 45.90 | 41.98 | 75.83 | 79.02 | 72.35 |
| Narrative | w/ (GNST-T5) | 61.46 | 66.49 | 55.35 | 44.30 | 46.23 | 42.08 | 76.20 | 79.35 | 72.76 |

Table 6: Models' performance on WebNLG test set, when fine-tuned with TEKGEN or GraphNarrative and further fine-tuned with WebNLG

| Model | BLEU | METEOR | chrF++ |
|---|---|---|---|
| T5-large | 50.38 | 39.98 | 68.06 |
| GN-T5 | 50.53 | 39.99 | 68.15 |
| GNST-T5 | 50.51 | 40.07 | 68.23 |

Table 7: Fine-tuning results on DART test set

| Model | ST | BLEU | METEOR | chrF++ |
|---|---|---|---|---|
| BART-large | w/o | 41.51 | 23.62 | 47.13 |
| BART-large | w/ | 48.32 | 29.90 | 57.50 |
| T5-large | w/o | 43.03 | 24.21 | 48.05 |
| T5-large | w/ | **49.83** | **30.52** | **58.25** |

Table 8: Performance of fine-tuning BART-large and T5-large on the TEKGEN dataset

**4) Ablation study of sentence trimming**. We demonstrate the effectiveness of sentence trimming in improving model performance on GraphNarrative, TEKGEN, WebNLG, and DART by fine-tuning PLMs with and without sentence trimming, respectively. (1) For GraphNarrative, we fine-tuned T5 and BART models using the setup described in Part 2, Section 6.3. (2) For TEKGEN, we fine-tuned the T5-large and BART-large models using the serialized triples from (Agarwal et al., 2021), with the same hyperparameters as in Part 2, Section 6.3. (3) For WebNLG and DART, we conducted zero-shot learning and fine-tuning experiments as described in Part 3, Section 6.3. (4) Additionally, on the WebNLG dataset, we carried out further fine-tuning of the T5-large model fine-tuned on TEKGEN in (2), applying the same hyperparameters as in Part 3, Section 6.3.

The results of (1) and (2) are in Tables 3 and 8. The metrics (BLEU, METEOR, chrF++) consistently improve with sentence trimming, further verifying the efficacy of sentence trimming. The results of (3) are in Tables 4, 6 and 7, and Ta-

ble 6 also shows the results of (4). In these results, the fine-tuned PLMs on GraphNarrative and TEKGEN with sentence trimming consistently outperformed their non-trimming counterparts. These findings underscore the effectiveness of sentence trimming in enhancing PLM performance. It is worth noting that, as Tables 6 and 7 show, on human-annotated WebNLG and DART the models did not gain much from sentence trimming after they are fine-tuned on these datasets. The main reason is that human-annotated datasets generally have well-aligned graph-text pairs and thus cannot be substantially improved by trimming.

**5) Sentence trimming in mitigating hallucination**. We randomly sampled 100 graphs from GraphNarrative test set, along with the corresponding sentences generated by GNST-T5 and GN-T5. We shuffled the 200 pairs and used three human evaluators to score the pairs, in the same fashion as in Part 1, Section 6.3. The results are in Table 9, which shows a reduction of 1.4 hallucinated entities and 1.0 hallucinated relations per instance from GN-T5 to GNST-T5, suggesting that sentence trimming effectively mitigates hallucinations. Furthermore, sentences generated by both models exhibit on average less than 0.07 missed entities and 0.38 missed relations per instance. Regarding grammar, sentences generated by GNST-T5 received

| ST | Hallucinated Entities | Missed Entities | Hallucinated Relations | Missed Relations | Grammar |
|---|---|---|---|---|---|
| w/o | 1.643 | 0.063 | 1.363 | 0.240 | 4.613 |
| w/ | 0.260 | 0.056 | 0.300 | 0.370 | 4.356 |

Table 9: Human evaluation of sentences generated by T5-large with and without sentence trimming

| Input Graph | Sentence Generated by GN-T5 | Sentence Generated by GNST-T5 |
|---|---|---|
| (Arthur Morry, place of death, Brisbane) | Arthur Morry died in Brisbane, Queensland, aged 79. | Arthur Morry died in Brisbane. |
| (Annapolis Maryland, location contains, US Naval Academy) | During World War II, the US Naval Academy in Annapolis, Maryland was renamed the US Naval Academy in Annapolis, Maryland, and the US Naval Academy in Annapolis, Maryland was renamed the US Naval Academy in Annapolis, Maryland. | US Naval Academy in Annapolis, Maryland. |
| (Goldie Gets Along, film directed by, Malcolm St. Clair filmmaker) (Goldie Gets Along, film performance actor, Lili Damita) (Goldie Gets Along, film performance actor, Charles Morton actor) | Goldie Gets Along is a 1951 American comedy film directed by Malcolm St. Clair (filmmaker) and starring Lili Damita and Charles Morton (actor). | Goldie Gets Along was directed by Malcolm St. Clair (filmmaker) and starred Lili Damita and Charles Morton (actor). |

Table 10: Comparison of generated sentences with and without sentence trimming for sample input graphs

slightly lower scores than GN-T5. Nevertheless, these scores remain acceptable, with on average less than one grammar error per instance.

Table 10 illustrates the sentences generated by GNST-T5 and GN-T5 for a few input graphs. GN-T5 tends to fabricate facts that are incorrect or non-existent in the real world (e.g., Arthur Morry's age of death, the renaming of the US Naval Academy, and Goldie Gets Along's year of release) or not present in input graphs (e.g., Goldie Gets Along's genre). In contrast, GNST-T5 generated fluent sentences without fabricating facts, barring a phrase instead of a complete sentence for the second example.

**6) Limitations of star graph datasets**. As explained in Section 1, existing large-scale datasets such as TEKGEN contain predominantly star graphs. We used GraphNarrative to investigate the limitations of star graph datasets.

More specifically, we separated the graph-sentence pairs in GraphNarrative into *star* instances (with star graphs) and *non-star* instances (without star graphs). We excluded instances with two or three entities, as they could be considered as both paths and stars. Table 11 provides the distributions of these two types of instances. The number of non-star instances in all three sets is approximately 3.5 times as many as the star instances. To help ensure a fair comparison, we randomly selected an equal number of non-star instances as the star instances for each of the three sets, e.g., there are 290,047 star graphs and the same number of non-star graphs in the training set of our prepared dataset.

Using the dataset prepared this way, we fine-tuned T5-large model with and without sentence trimming for 10 epochs under early stopping patience 5, using the same other hyperparameters as in Part 2, Section 6.3. The results are in Table 12. Across the board, models trained using star instances exhibited the highest performance when tested using star instances too, and similarly regarding non-star instances. Furthermore, models trained on non-star instances and tested on star instances tended to outperform models trained on star

instances and tested on non-star instances. These results indicate that a PLM fine-tuned on a dataset consisting solely of star graphs performs poorly when applied to general graph shapes, which are commonly encountered in real-world applications. Fine-tuning PLMs on diverse graph shapes enhances their generalization capability.

| | Star | Non-star | All |
|---|---|---|---|
| train | 290,047 | 1,022,303 | 1,312,350 |
| dev | 16,192 | 56,612 | 72,804 |
| test | 16,425 | 58,517 | 74,942 |
| all | 322,664 | 1,137,432 | 1,460,096 |

Table 11: Number of star and non-star instances in GraphNarrative

| ST | Train | Test | BLUE | METEOR | chrF++ |
|---|---|---|---|---|---|
| w/o | star | star | 36.57 | 22.75 | 46.70 |
| | | non-star | 30.64 | 21.89 | 45.48 |
| | | both | 33.54 | 22.32 | 46.09 |
| | non-star | star | 34.02 | 21.67 | 44.54 |
| | | non-star | 37.18 | 23.99 | 50.02 |
| | | both | 35.71 | 22.99 | 47.57 |
| | both | star | 36.23 | 22.94 | 47.06 |
| | | non-star | 36.86 | 24.32 | 50.61 |
| | | both | 36.56 | 23.63 | 48.83 |
| w/ | star | star | 47.70 | 26.62 | 52.08 |
| | | non-star | 37.75 | 25.17 | 50.20 |
| | | both | 42.48 | 25.88 | 51.14 |
| | non-star | star | 45.83 | 25.72 | 50.33 |
| | | non-star | 47.30 | 27.73 | 55.21 |
| | | both | 46.61 | 26.73 | 52.77 |
| | both | star | 47.60 | 26.87 | 52.52 |
| | | non-star | 46.83 | 27.89 | 55.45 |
| | | both | 47.20 | 27.38 | 53.99 |

Table 12: Model performance, star vs. non-star graphs

# 7 Conclusion

In this paper, we proposed a novel approach to mitigating hallucination in natural language generation from large-scale, open-domain knowledge graphs. We released a large graph-to-text dataset with diverse graph shapes that fills the gap between existing datasets and real-world settings. The experiment results show the effectiveness of our hallucination mitigation approach as well as the usefulness of the dataset.

## Limitations

1) The creation of GraphNarrative and the sentence trimming method leverage an existing mapping between the knowledge graph entities and Wikipedia entities. Given other text corpora and knowledge graphs, creating such a mapping is a non-trivial undertaking that often requires named entity recognition and disambiguation techniques. 2) The sentence trimming approach may introduce grammatical errors into generated sentences. 3) The method focuses on describing the content of an input graph only, without considering context information such as neighboring entities in the knowledge graph. Such extra information may be preferred by a user given certain application contexts or may make the input graph's narration more natural. 4) The creation of GraphNarrative does not consider multiary relationships in knowledge graphs. More specifically, the Freebase used in our work is a version in which multiary relationships were converted into binary relationships (Shirvani-Mahdavi et al., 2023). In general, there is a lack of inquiry into multiary relationships in graph-to-text models. To the best of our knowledge, the only work in this area that discusses such multiary relationships is (Agarwal et al., 2021) and they also converted multiary relationships into binary ones. 5) A couple of studies (Agarwal et al., 2021; Wang et al., 2021) attempted to address hallucination by further fine-tuning PLMs on WebNLG after fine-tuning on noisier automatically-extracted datasets. It will be informative to conduct a human evaluation comparison between their approaches and the sentence trimming method proposed in our work. Similarly, our future work includes a human evaluation comparison with the filtering-based method (Ma et al., 2022) which we empirically compared with in Appendix C.1. 6) The sentence trimming algorithm only removes irrelevant portions from the beginning and the end of a sentence, leaving the token sequence in the middle intact. It is possible the middle portion also contains tokens irrelevant to the input graph.

## Ethics Statement

In the course of conducting our research, we have striven to remain aware and attentive to potential ethical implications and challenges. Our work was informed by the following ethical considerations.

*Ethical use of generated content*. Given that our research focuses on producing natural language descriptions of knowledge graphs, we are particularly aware of the potential misuse of our method for the generation of false, deceptive, biased or unfair contents. Particularly, our sentence trimming method aims to minimize such potential misuse by aiding in reducing hallucinations.

We also recognize that natural language descriptions generated using our dataset and algorithm can be repurposed in various ways. We firmly urge users and developers to use this content responsibly, particularly with respect to intellectual property rights. Furthermore, we recommend users clearly label AI-generated content, promoting transparency and trust.

*Data privacy and bias*. Our GraphNarrative dataset uses publicly available data, particularly Freebase and Wikipedia, which do not contain information that violates anyone's privacy to the best of our knowledge.

Our reliance on Wikipedia may inadvertently introduce bias, as Wikipedia content can reflect the views of its contributors. We are also aware this potential bias could be more intense in less commonly spoken languages, where the number of contributors might be limited.

## Acknowledgments

This work is supported by the National Science Foundation under Grants 1719054, 1937143, and 2333834. We extend our gratitude to Dr. Won Hwa Kim and Xin Ma for provisioning of computational resources in supporting this work. Special thanks to Nasim Shirvani-Mahdavi and Haiqi Zhang for their vital contributions to the human evaluation process, to Yogesh Gurjar for resolving a bug in our experiment scripts, and to Mohammed Samiul Saeef for his guidance and expertise on using Freebase data graph.

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

---

**Algorithm 1:** Sentence Trimming

---

**Input:** $W$: A co-reference resolved Wikipedia sentence; $G$: The graph for $W$ based on graph-text alignment

**Output:** $W_{trim}$: The trimmed text sequence

1   $M \leftarrow \{\}$
2   **foreach** $(s, p, o) \in G$ **do**
3     $s' \leftarrow s.remove(special\_tokens)$
4     $o' \leftarrow o.remove(special\_tokens)$
5     $W \leftarrow W.replace((s, o), (s', o'))$
6     $M[s'], M[o'] \leftarrow s, o$
7   $W_{tree} \leftarrow W.dependency\_parsing()$
8   $min\_pos, max\_pos \leftarrow W.length, 0$
9   **foreach** $(s, p, o) \in G$ **do**
10    $sdp \leftarrow shortest\_path(W_{tree}, s', o')$
11    **foreach** $node \in sdp$ **do**
12      $min\_pos \leftarrow \min(min\_pos, node.start)$
13      $max\_pos \leftarrow \max(max\_pos, node.end)$
14   $W_{trim} \leftarrow W[min\_pos : max\_pos]$
15   **foreach** $k \in M.keys()$ **do**
16    $W_{trim} \leftarrow W_{trim}.replace(k, M[k])$
17   **return** $W_{trim}$

---

## A   The Sentence Trimming Algorithm

In Algorithm 1, Lines 1–6 are to ensure that an entity consisting of multiple tokens is tokenized into one single token, and the mapping $M$ is for recovering the entity's tokens from $W$ in producing $W_{trim}$ (Lines 15–16). Lines 9–14 find the leftmost position $min\_pos$ and the rightmost position $max\_pos$ from $W$ by scanning each triple $(s, p, o)$ in $G$ and finding the tokens on the corresponding SDPs, as explained in Section 5. The variable $node$ denotes a token on the SDP in $W_{tree}$ between entities $s$ and $o$. $node.start$ and $node.end$ denote $node$'s starting position index and ending position index in $W$, respectively. $node.start$, $node.end$, $min\_pos$, and $max\_pos$ are on character level. If $node$ appears multiple times in $W$, $node.start$ will be the start index of the first one and $node.end$ will be the end index of the last one.

## B   More Details about GraphNarrative

### B.1   List of Domains

*Seen domains.*   location, people, sports, music, government, organization, education, film, tv, book, award, military, soccer, time, geography, business, olympics, transportation, broadcast, baseball, fictional_universe, biology, influence, language, computer, cvg, aviation, architecture, protected_sites, religion, symbols, travel, visual_art, basketball, royalty, astronomy, american_football, metropolitan_transit, comic_books, law, media_common, spaceflight, tennis, boats, medicine, meteorology, automotive, theater, internet, amusement_parks, event, comic_strips, measurement_unit, finance, radio, physics

*Unseen domains.*   martial_arts, games, ice_hockey, cricket, rail, food, opera, projects, dining, skiing, conferences, library, exhibitions, zoos, boxing, engineering, digicams, venture_capital, chemistry, celebrities, chess, interests, distilled_spirits, comedy, fashion, geology, wine, bicycles

### B.2   GraphNarrative **Characteristics**

*Graph distribution.*   The distributions of graphs in GraphNarrative by numbers of triples and entities are shown in base-10 logarithmic scale in Figure 4 and Figure 5, respectively. Furthermore, the distribution of distinct GraphNarrative graph shapes by number of entities is in Table 13.

*Text distribution.* The dataset exhibits an average sentence length of 34.68 tokens for original sen-

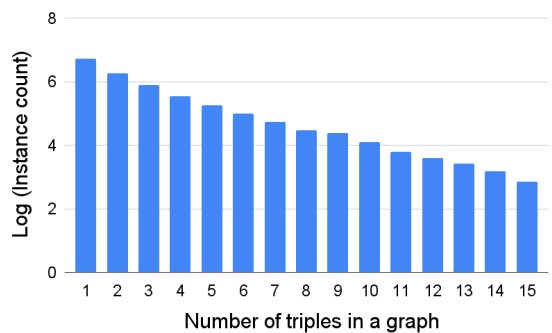

Figure 4: Distribution of GraphNarrative instances by number of triples in graphs

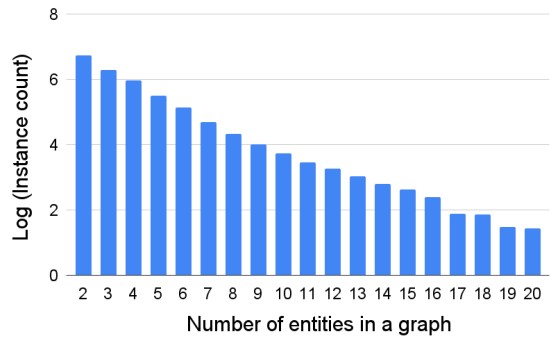

Figure 5: Distribution of GraphNarrative instances by number of entities in graphs

| #entities | 2 | 3 | 4 | 5 | 6 | 7 | 8 | 9 | 10 | 11 |
|-----------|---|---|---|----|----|-----|-----|------|------|-----|
| #shapes | 1 | 2 | 7 | 23 | 122 | 705 | 1690 | 1705 | 1267 | 830 |

| #entities | 12 | 13 | 14 | 15 | 16 | 17 | 18 | 19 | 20 | all |
|-----------|----|----|----|----|----|----|----|----|----|-----|
| #shapes | 542 | 378 | 222 | 176 | 106 | 58 | 52 | 22 | 12 | 7920 |

Table 13: Distribution of distinct GraphNarrative graph shapes by number of entities

| #Tokens | Original Sentence | Trimmed Sentence |
|---------|-------------------|------------------|
| 0-10 | 64,861 (0.74%) | 1,734,759 (19.78%) |
| 10-20 | 1,442,582 (16.45%) | 3,369,747 (38.43%) |
| 20-30 | 2,592,651 (29.56%) | 1,944,225 (22.17%) |
| 30-40 | 2,048,174 (23.36%) | 919,554 (10.49%) |
| 40-50 | 1,215,928 (13.87%) | 416,187 (4.75%) |
| 50-60 | 649,403 (7.41%) | 190,363 (2.17%) |
| 60-70 | 339,671 (3.87%) | 91,289 (1.04%) |
| 70-80 | 181,699 (2.07%) | 45,38 (0.52%) |
| 80-90 | 97,618 (1.11%) | 23,823 (0.27%) |
| 90-100 | 53,981 (0.62%) | 13,299 (0.15%) |
| 100-110 | 83,066 (0.95%) | 21,008 (0.24%) |

Table 14: Distribution of GraphNarrative instances by sentence length

| #Triples | #Tokens | #Triples | #Tokens | #Triples | #Tokens |
|----------|---------|----------|---------|----------|---------|
| 1 | 14.15 | 2 | 20.32 | 3 | 23.12 |
| 4 | 26.29 | 5 | 28.21 | 6 | 29.74 |
| 7 | 31.12 | 8 | 33.41 | 9 | 30.39 |
| 10 | 35.90 | 11 | 39.53 | 12 | 42.49 |
| 13 | 43.60 | 14 | 49.71 | 15 | 50.21 |

Table 15: Average GraphNarrative sentence length by number of triples in graphs

tences and 20.66 tokens for trimmed sentences. Table 14 provides detailed distribution of sentence lengths. Table 15 presents the average sentence token counts by number of triples in the graphs. It underscores that our model was trained using a diverse set of examples, including those with lengthy sentences and a substantial number of triples.

### B.3 Dataset Creation

#### B.3.1 Pre-processing of text corpus and knowledge graph

Our text corpus is derived from the Wikipedia dump [3] released on Sep. 1st, 2019. In preprocessing the corpus, we used WikiExtractor (Attardi, 2019) to transform the raw Wikipedia dump in compressed XML file into numerous plain text files containing bodies of Wikipedia articles without tables, infoboxes, table of contents, categories, and so on. [4]

Our Freebase knowledge graph is from (Shirvani-Mahdavi et al., 2023) which used the most recent Freebase dump [5] as the data source. Most relations in the graph form semantically-redundant re-

verse pairs. If the input graph triples to a graph-to-text model containing such reverse edges, we only need to simply retain one edge out of each redundant pair. Hence, we did exactly that in pre-processing the whole Freebase dump so that our input graphs have no reverse edges. Furthermore, our pre-processing also removed the mediator (CVT) nodes (Bollacker et al., 2008) by concatenating edges connected through mediator nodes.

#### B.3.2 Graph-text alignment

*Wikipedia-to-Freebase entity mapping.* We collected a Wikipedia-to-Freebase entity mapping between 4,408,115 English Wikipedia titles and their corresponding Freebase entities. The mapping was created by employing three methods, as follows. 1) Parsing the Freebase data dump to obtain a Wikipedia-to-Freebase entity mapping using https://github.com/saleiro/Freebase-to-Wikipedia. 2) Inferring from a Wikipedia-to-Wikidata mapping in wikimapper (Klie, 2022) and a Wikidata-to-Freebase mapping at https://developers.google.com/freebase. 3) Inferring from the Wikipedia-to-DBpedia and the DBpedia-to-Freebase mappings at http://downloads.dbpedia.org/2016-10/

---

[3] https://dumps.wikimedia.org
[4] https://en.wikipedia.org/wiki/Wikipedia:Manual_of_Style/Layout
[5] https://developers.google.com/freebase

| Threshold | Dev (w/o ST) | Dev (w/ ST) | Test (w/o ST) | Test (w/ ST) | Train (w/o ST) | Train (w/ ST) | Total (w/o ST) | Total (w/ ST) |
|---|---|---|---|---|---|---|---|---|
| 0.8 | 74 (0.0169%) | 791 (0.1806%) | 65 (0.0144%) | 931 (0.206%) | 1328 (0.0168%) | 14787 (0.1877%) | 1467 (0.017%) | 16509 (0.1883%) |
| 0.5 | 14952 (3.416%) | 71565 (15.357%) | 15843 (3.507%) | 75445 (16.689%) | 269541 (3.418%) | 1293864 (16.405%) | 300336 (3.43%) | 1382074 (15.76%) |
| 0.3 | 142974 (32.678%) | 256357 (58.593%) | 148666 (32.892%) | 266094 (58.886%) | 2579921 (32.740%) | 4621448 (58.659%) | 4970561 (56.68%) | 7445899 (84.88%) |

Table 16: Number of remaining instances after filtering using different thresholds of ROUGE-1 similarity scores

| Dataset | Entities | Triples | Relations | Star Graphs | Shapes |
|---|---|---|---|---|---|
| Filtered | 307,590 | 437,519 | 974 | 47% | 597 |
| GraphNarrative | 1,853,752 | 15,472,249 | 1,724 | 22% | 7,920 |

Table 17: Statistics of GraphNarrative and its filtered dataset

core-i18n/en/. The overall Wikipedia-to-Freebase entity mapping is obtained by combining all three methods and eliminating conflicting entity mappings. The mapping file link can be found at https://github.com/idirlab/graphnarrator.

*Coreference resolution.* To produce more graph-text pairs for GraphNarrative, we used AllenNLP's coreference resolution (Gardner et al., 2017; Lee et al., 2017) in default settings to replace Wikipedia token spans with the entities they refer to. We conducted human evaluation to assess the quality of the coreference resolution results on 20 randomly selected Wikipedia articles. The assessment yielded a precision of 91.4% (630 of the 689 resolved entity coreferences were correct) and a recall of 98.3% (11 entity coreferences were missed).

| #Triples | BLEU (GN-T5) | BLEU (GNST-T5) |
|---|---|---|
| 1 | 13.60 | 25.97 |
| 2 | 19.90 | 27.53 |
| 3 | 24.07 | 29.73 |
| 4 | 30.51 | 32.62 |
| 5 | 32.95 | 35.22 |
| 6 | 38.81 | 39.73 |
| 7 | 42.74 | 41.55 |
| 8 | 42.23 | 42.10 |
| 9 | 55.73 | 51.83 |
| 10 | 45.84 | 48.08 |
| 11 | 41.72 | 42.71 |
| 12 | 38.09 | 39.33 |
| 13 | 41.77 | 43.27 |
| 14 | 33.47 | 36.95 |
| 15 | 34.04 | 38.44 |

Table 18: Distribution of GNST-T5 and GN-T5 model performance in BLEU scores on GraphNarrative test set

## C   Additional Experiments

### C.1   Comparing sentence trimming with filtering

We compared sentence trimming with a similar but different filtering method proposed in (Ma et al., 2022). Their method also aimed to reduce disparities in datasets as a way of mitigating hallucination. However, different from our approach which aligns sentences better with input graphs by trimming away portions of sentences, the filtering method removes graph-text pairs from the DART dataset where the ROUGE-1 similarity score between the graph and the text is below 0.8.

We applied the same filtering method on GraphNarrative. Table 16 provides a breakdown of the remaining instances after filtering using different thresholds. A relatively low threshold of 0.3 removed 43.32% of the instances in GraphNarrative. When we raised the threshold to 0.8, almost all instances were eliminated. In comparison, the threshold of 0.8 applied on DART allowed for retaining 88% of its instances. This is because the human-annotated DART has well-aligned graph-text pairs.

We compared sentence trimming with filtering using the 269,541 instances left in the training set and 71,565 in the development set, under threshold 0.5. We fine-tuned the T5-large model for 10 epochs with early stopping patience 5, using the same other hyperparameters as on the full dataset. The number of training steps is different from the full dataset because this subset is about 30 times smaller. We used early stopping to avoid overfitting. Then we used the resulting model, which we call Filter-T5, for zero-shot prediction on WebNLG and DART test sets. The results are shown in Table 4. GNST-T5 slightly outperformed Filter-T5. To understand this, we compared the statistics of the filtered dataset and the full dataset (and thus the dataset after sentence trimming since trimming does not alter the graphs in the dataset), as in Table 17. The filtered dataset exhibits a significant reduction in size and diversity in terms of number of distinct entities, relations, triples and shapes. We conjecture that this contributes to its performance degeneration in comparison with GNST-T5.

### C.2   Performance of GNST-T5 and GN-T5 by input size

Table 18 shows the performance of GNST-T5 and GN-T5 in BLEU scores on graphs of varying sizes, i.e., number of triples. The results help gauge whether the models generalize well for long inputs. Notably, the performance of both models on extended inputs is better than or on par with their performance on shorter inputs.

## D The 41 Shapes in WebNLG

Figure 6 displays the distinct graph shapes in the WebNLG dataset, in descending order by number of instances.

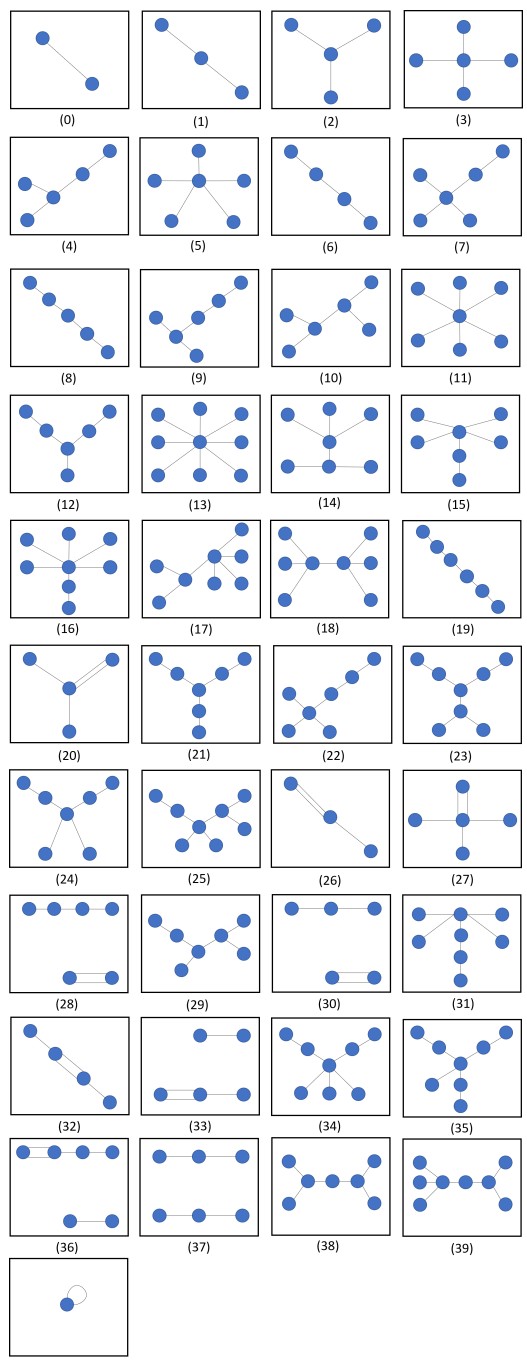

Figure 6: WebNLG shapes