# OpenReview forum: "Hallucination Mitigation in Natural Language Generation from Large-Scale Open-Domain Knowledge Graphs"
_EMNLP/2023/Conference — EMNLP 2023 Main_

### Official Review · Reviewer_hPGF · 2023-07-30

**Soundness:** 3

**Excitement:**

3: Ambivalent: It has merits (e.g., it reports state-of-the-art results, the idea is nice), but there are key weaknesses (e.g., it describes incremental work), and it can significantly benefit from another round of revision. However, I won't object to accepting it if my co-reviewers champion it.

**Missing References:**

Linyong Nan, Dragomir Radev, Rui Zhang, Amrit Rau, Abhinand Sivaprasad, Chiachun Hsieh, Xiangru Tang, Aadit Vyas, Neha Verma, Pranav Krishna, Yangxiaokang Liu, Nadia Irwanto, Jessica Pan, Faiaz Rahman, Ahmad Zaidi, Mutethia Mutuma, Yasin Tarabar, Ankit Gupta, Tao Yu, et al.. 2021. DART: Open-Domain Structured Data Record to Text Generation. In Proceedings of the 2021 Conference of the North American Chapter of the Association for Computational Linguistics: Human Language Technologies, pages 432–447, Online. Association for Computational Linguistics.

Kaixin Ma, Hao Cheng, Xiaodong Liu, Eric Nyberg, and Jianfeng Gao. 2022. Open Domain Question Answering with A Unified Knowledge Interface. In Proceedings of the 60th Annual Meeting of the Association for Computational Linguistics (Volume 1: Long Papers), pages 1605–1620, Dublin, Ireland. Association for Computational Linguistics.

**Paper Topic And Main Contributions:**

This paper introduced a new dataset, GraphNarrative, for training graph-to-text generation models. The authors first collected Wikipedia sentences, and then find the corresponding entities in Freebase for every entity mention in the sentences. Then the knowledge triples from Freebase are further grounded to those sentences with lexical matching. The statistics show that the resulting graph-to-sentence dataset contains a much less percentage of star graphs (one central entity surrounded by other relations) compare to previous datasets. To reduce the misalignment between the graph and the sentence, the authors also proposed a method to trim sentences. They leverage the dependency graph of the sentence and try to find the shortest paths within the graph that covers all entities from the paired freebase triples.
The authors experimented with BART and T5 models, where they first finetune the models on GraphNarrative and then finetune on the WebNLG dataset. They also specifically compared finetuning models on original sentences vs trimmed sentences. The results show that finetuning on trimmed sentences leads to higher output quality and reduced hallucinations.

**Questions For The Authors:**

What is the average length of the sentences in your dataset? What is the sentence length distribution? If the sentences are mostly short, then the models trained on this dataset might not generalize well to tasks with long inputs.

**Reasons To Accept:**

1. The proposed dataset could be useful for future research for training better data-to-text generation models. The proposed method could be applied to other textual corpus and knowledge graphs.
2. The automatic evaluation shows that pretraining on GraphNarrative and then finetuning on WebNLG effectively improved the model’s performance. The human evaluation shows that the proposed sentence trimming method effectively reduced the hallucination of the models.

**Reasons To Reject:**

1. The generalization of the proposed dataset is not well studied. For example, GraphNarrative is more than 300x larger than WebNLG dataset, and it covers a lot more graph types, entities, relations and domains than WebNLG. Intuitively, models trained on GraphNarrative should be able to generalize well to WebNLG without further finetuning. However, no zero-shot results are presented.
2. The necessity of applying the sentence trimming method is not well established. For example, in Table 3 and Table 4, the results show that training on the trimmed sentences leads to more grammatical errors in the outputs, suggesting that trimming could lead to unnatural sentences. What if you simply discard the sentences that do not align well with the graph? Given the large size of GraphNarrative, there should be a decent amount of instances left after pruning. The authors should also consider experimenting with different amounts of training data to examine the effect of training size.
3. The comparison with recent related studies is lacking: for example, Nan et al. 21 proposed the DART dataset, which is used for studying data-to-text generation. In fact, DART could be another evaluation set for understanding the generalization of GraphNarrative. Ma et al. 22 proposed a data-to-text generation model. They also proposed to filter out instances from training data that might lead to hallucinations or missing information. The authors should consider discussing and comparing with these related works.

**Reproducibility:**

4: Could mostly reproduce the results, but there may be some variation because of sample variance or minor variations in their interpretation of the protocol or method.

**Reviewer Confidence:**

5: Positive that my evaluation is correct. I read the paper very carefully and I am very familiar with related work.

---

> ### Author Rebuttal · Authors · 2023-08-29
>
> We appreciate the valuable review and suggestion which have helped us conduct more experiments that will be included in the camera-ready version of the paper. Thank you!
>
>
> **1.** “The generalization of the proposed dataset is not well studied. For example, GraphNarrative is more than 300x larger than WebNLG dataset, and it covers a lot more graph types, entities, relations and domains than WebNLG. Intuitively, models trained on GraphNarrative should be able to generalize well to WebNLG without further finetuning. However, no zero-shot results are presented.”
>
> **Response**
>
> Thank you so much for the comment, based on which we conducted three zero-shot experiments on the WebNLG dataset: 1) using original graph-to-text task T5-large model (Ribeiro et al., 2021) to directly perform inference, without fine-tunning on WebNLG; 2) T5-large fine-tuned on GraphNarrative without sentence trimming (GN-T5); and 3)T5-large fine-tuned on GraphNarrative with sentence trimming (GNST-T5). The three experiments used the same linearization method for the graphs and parameters mentioned in the submission.  The results are shown below, and we will discuss these results in the camera-ready version.
>
> [Ribeiro et al., 2021] Leonardo FR Ribeiro, Martin Schmitt, Hinrich Schütze, and Iryna Gurevych. 2021. Investigating pretrained language models for graph-to-text generation. In Proceedings of the 3rd Workshop on Natural Language Processing for Conversational AI, pages 211–227.
>
> | Model    | BLEU  | METEOR | ChrF++ |
> | -------- | ----- | ------ | ------ |
> | T5-large | 4.01  | 9.54   | 24.64  |
> | GN-T5    | 21.38 | 31.82  | 56.83  |
> | GNST-T5  | 27.60 | 32.27  | 56.81  |
>
> As the results show, the zero-shot performance of GN-T5 exceeds the original T5-large model by a large margin, indicating the GraphNarrative dataset improves the generalization ability of PLMs. GNST-T5 further improves the model’s performance, reflecting the effect of sentence trimming. We will also include results for more models (e.g., BART-based, GPT-based) in the camera-ready version.
>
>
>
> The zero-shot results of GN-T5 and GNST-T5 shown in the table are worse than the results from T5 models fine-tuned using  WebNLG (Table 7 in the submission). GNST-T5 and GN-T5 were not fine-tuned using WebNLG, and thus it is expected for a model fine-tuned using WebNLG to perform better when both models are compared on their inference accuracy measured by WebNLG. The reasons could be 1) the underlying knowledge graphs are different---WebNLG was generated using DBPedia as the underlying knowledge graph while GraphNarrative was generated using Freebase. 2) WebNLG is human-annotated, with relatively good alignment between graphs and texts, while GraphNarrative was automatically collected and thus has more misalignments. 3) the differences in evaluation scores could also be due to the more diverse expressions in Wikipedia sentences (in GraphNarrative) compared to those crafted by human annotators (in WebNLG). Automatic evaluation scores such as BLEU rely on n-gram similarity between predicted and ground-truth sentences. Consider a single-triple graph. A human annotator might use one particular expression, while the model generates an alternative expression using different wording. In such cases, even if the model's expression is accurate, the evaluation score could be considerably low.
>
>
> **2.** “The necessity of applying the sentence trimming method is not well established. For example, in Table 3 and Table 4, the results show that training on the trimmed sentences leads to more grammatical errors in the outputs, suggesting that trimming could lead to unnatural sentences. What if you simply discard the sentences that do not align well with the graph? Given the large size of GraphNarrative, there should be a decent amount of instances left after pruning. The authors should also consider experimenting with different amounts of training data to examine the effect of training size.”
>
>
>
> **Response**
>
>
>
> **2.1** “What if you simply discard the sentences that do not align well with the graph? Given the large 	size of GraphNarrative, there should be a decent amount of instances left after pruning.”
>
>
>
> Two categories of misalignment can be discerned: entity misalignment and relationship misalignment.
>
> In the context of entity misalignment, instances where certain entities within a graph fail to be referenced in the corresponding sentence have been deliberately omitted from the GraphNarrative dataset. It's important to note that it's quite common for sentences to encompass more entities than those present in the graph, given the inherent incompleteness of real-world knowledge graphs. Implementing such exclusion criteria would likely result in the removal of a substantial portion of instances.
>
>
>
> Regarding relationship misalignment, the challenge lies in accurately determining whether the relationships depicted in the graph are appropriately aligned with the sentence. The diversity of natural language expressions compounds this challenge, making it difficult to establish precise alignment.
>
> We extend our gratitude for your valuable suggestion to utilize the filter method as described in (Ma et al. 2022). We have diligently applied this methodology to our dataset and will elaborate on our findings in our subsequent response.
>
>
>
>
>
> **2.2** “The necessity of applying the sentence trimming method is not well established. For example, in Table 3 and Table 4, the results show that training on the trimmed sentences leads to more grammatical errors in the outputs, suggesting that trimming could lead to unnatural sentences.
>
>
>
> It's important to note that, while grammar errors slightly increase after applying sentence trimming, the reduction in hallucinations is much more significant. As Table 4 shows, with sentence trimming, numbers of hallucinated entities and hallucinated relations were reduced by 1.4 and 1.0, respectively, while grammar score only decreased from 4.613 to 4.356 (a score of 5 means no grammatical error, 4 means one error, and so on). This trade-off is especially relevant in applications where truthfulness and semantic coherence hold greater importance than grammar. Furthermore, we are actively engaged in designing methods to mitigate grammar errors.
>
>
>
> **2.3** "The authors should also consider experimenting with different amounts of training data to examine the effect of training size.”
>
>
>
> Thank you for the suggestion. We will report the results of such experiments in the camera-ready version.
>
>
>
> **3.** “The comparison with recent related studies is lacking: for example, Nan et al. 21 proposed the DART dataset, which is used for studying data-to-text generation. In fact, DART could be another evaluation set for understanding the generalization of GraphNarrative. (Ma et al. 2022) proposed a data-to-text generation model. They also proposed to filter out instances from training data that might lead to hallucinations or missing information. The authors should consider discussing and comparing with these related works.”
>
>
>
> **Response**
>
>
>
> We will include the discussion of the two papers in the camera-ready version. Based on the comment, 	we obtained statistics of the DART dataset and did additional experiments of zero-shot learning and fine-tuning the original T5 large, GN-T5, and GNST-T5 on DART. We also applied the filtering method by (Ma et al., 2022). Below we summarize the results, and we will include details in the paper.
>
>
>
> [Ma et al., 2022] Kaixin Ma, Hao Cheng, Xiaodong Liu, Eric Nyberg, and Jianfeng Gao. 2022. Open Domain Question Answering with A Unified Knowledge Interface. In Proceedings of the 60th Annual Meeting of the Association for Computational Linguistics (Volume 1: Long Papers), pages 1605–1620, Dublin, Ireland.
>
>
>
>
>
> **3.1** The distinctions between DART and the GraphNarrative dataset are as follows:
>
>
>
> Graph characteristics: A notable disparity is observed in the graph structures. Specifically, a significant majority (83%) of the graphs in DART take the form of star graphs. An overview of its shape distribution can be found in the table below.
>
>
>
> Dataset nature: DART is primarily a human-annotated dataset. It integrates data from various sources including Wiki tables (human-annotated descriptions), WebNLG (also human-annotated), and E2E (human annotated too). In contrast, GraphNarrative is an automatically collected dataset.
>
>
>
> Application focus: DART predominantly is for table-to-text generation. Notably, it lacks a dedicated, large-scale, open-domain knowledge graph akin to Wikidata, DBpedia, Freebase, and similar resources. This aspect diverges from our specific aim: the automated generation of natural language descriptions for subgraphs sourced from a vast, open-domain knowledge graph.
>
>
>
>
>
> |         | **Instances** | **Star Instance** | **[TABLECONTEXT] instance** | **Source counter**                                                                                                                                      |
> | ------- | ------------- | ----------------- | --------------------------- | ------------------------------------------------------------------------------------------------------------------------------------------------------- |
> | Dev     | 2768          | 2484              | 61                          | {'e2e': 1475, 'webnlg': 872, 'WikiSQL_decl_sents': 249, 'WikiTableQuestions_lily': 106, 'WikiTableQuestions_mturk': 39, 'WikiSQL_lily': 27}             |
> | Test    | 5097          | 4308              | 69                          | {'webnlg': 2753, 'e2e': 1844, 'WikiSQL_decl_sents': 346, 'WikiTableQuestions_lily': 95, 'WikiTableQuestions_mturk': 34, 'WikiSQL_lily': 25}             |
> | Train   | 30526         | 24976             | 3176                        | {'e2e': 12530, 'webnlg': 6940, 'WikiTableQuestions_lily': 4700, 'WikiSQL_decl_sents': 3589, 'WikiTableQuestions_mturk': 2047, 'WikiSQL_lily': 720}      |
> | **Sum** | **38391**     | **31768**         | **3306**                    | **{'e2e': 15849, 'webnlg': 10565, 'WikiSQL_decl_sents': 4084, 'WikiTableQuestions_lily': 4901, 'WikiTableQuestions_mturk': 2120, 'WikiSQL_lily': 772}** |
>
>
>
>
>
> **3.2** Tables below show the model performance of zero-shot learning and fine-tuning on DART. For fine-tuning, we trained for 100 epochs with early-stopping patience 20, and we picked the best performance model on development set, with linearization and other hyperparameters set the same as in .
>
>
>
> Zero-shot results:
>
>
>
> | Model    | BLEU  | METEOR | ChrF++ |
> | -------- | ----- | ------ | ------ |
> | T5-large | 3.44  | 7.93   | 23.17  |
> | GN-T5    | 19.35 | 27.35  | 50.41  |
> | GNST-T5  | 19.42 | 28.07  | 50.96  |
>
>
>
> Fine-tuning results:
>
>
>
> | Model    | BLEU  | METEOR | ChrF++ |
> | -------- | ----- | ------ | ------ |
> | T5-large | 50.38 | 39.98  | 68.06  |
> | GN-T5    | 50.53 | 39.99  | 68.15  |
> | GNST-T5  | 50.51 | 40.07  | 68.23  |
>
>
>
> The zero-shot results show that fine-tuning GraphNarrative improves the models’ generalization ability by a large margin.
>
>
>
> When fine-tuning DART, GN-T5 and GNST-T5 does not improve over T5-large much, similar to the results on WebNLG (Table 7 of submission). This is because both WebNLG and DART are human-annotated datasets without many hallucinations in their training dataset. As mentioned in Lines 176-180 of the paper, the human-annotated dataset is not realistic for a large-scale, open-domain setting. That is why we need the sentence trimming method, of which the effectiveness was demonstrated on both TEKGEN and GraphNarrative datasets.
>
>
>
> **3.3** We applied the filter method in (Ma et al. 2022) by filtering out instances in GraphNarrative dataset whose ROUGE-1 scores between graph and text fall below a threshold. We used three different thresholds: 0.8 (used by (Ma et al., 2022), based on communication with the authors of (Ma et al. 2022), 0.5, and 0.3. The numbers and percentages of instances above various threasholds are shown below.
>
>
>
> | Threshold | Dev (w/o trimming)| Dev (w/ trimming) | Test (w/o trimming) | Test (w/ trimming) | Train (w/o trimming) | Train (w/ trimming) | Total (w/o trimming) | Total (w/ trimming) |
> |-----------|-------------------|-------------------|---------------------|--------------------|----------------------|---------------------|----------------------|---------------------|
> | 0.8       | 74 (0.0169%)      | 791 (0.1806%)     | 65 (0.0144%)        | 931 (0.206%)       | 1328 (0.0168%)       | 14787 (0.1877%)     | 1467 (0.017%)        | 16509 (0.1883%)     |
> | 0.5       | 14952 (3.416%)    | 71565 (15.357%)   | 15843 (3.507%)      | 75445 (16.689%)    | 269541 (3.418%)      | 1293864 (16.405%)   | 300336 (3.43%)       | 1382074 (15.76%)    |
> | 0.3       | 142974 (32.678%)  | 256357 (58.593%)  | 148666 (32.892%)    | 266094 (58.886%)   | 2579921 (32.740%)    | 4621448 (58.659%)   | 4970561 (56.68%)     | 7445899 (84.88%)    |
>
>
>
> Half of the instances in our dataset were filtered out even with a low threhold of 0.3. Given 0.8 as the threshold, 99% of the instances were filtered out. The filtering method is not quite applicable to our dataset. DART essentially is a human-annotated dataset, and thus it is more clean than automatically collected dataset. After filtering out instances below score 0.8, there are still a large proportion of the instances (88%) remaining. A real-world knowledge graph is usually far from complete, as there could be many misalignmenst between the graph and text. By sentence trimming, we improve the ROUGE-1 score between graph and text, as can be seen by comparing the columns with and without sentence trimming in the table above.
>
>
>
>
>
>
>
> **4.** “What is the average length of the sentences in your dataset? What is the sentence length distribution? If the sentences are mostly short, then the models trained on this dataset might not generalize well to tasks with long inputs.”
>
>
>
> **Response**
>
>
>
> The dataset exhibits an average sentence length of 34.68 tokens for original sentences and 20.66 tokens for trimmed sentences. Additionally, the distribution of sentence lengths within our dataset is provided in the table below for reference. The majority of our dataset falls in the number of tokens from 10-40. Comparatively, DART boasts an average sentence length of 21.10 tokens, while WebNLG registers an average of 22.50 tokens. (Colas, Anthony, et al., 2023) Notably, the sentences within our dataset are not much shorter than those found in other comparable datasets.
>
>
>
> | #tokens | original sentence  | trimmed sentence   |
> | ------- | ------------------ | ------------------ |
> | 0-10    | 64.861k (0.74%)    | 1734.759k (19.78%) |
> | 10-20   | 1442.582k (16.45%) | 3369.747k (38.43%) |
> | 20-30   | 2592.651k (29.56%) | 1944.225k (22.17%) |
> | 30-40   | 2048.174k (23.36%) | 919.554k (10.49%)  |
> | 40-50   | 1215.928k (13.87%) | 416.187k (4.75%)   |
> | 50-60   | 649.403k (7.41%)   | 190.363k (2.17%)   |
> | 60-70   | 339.671k (3.87%)   | 91.289k (1.04%)    |
> | 70-80   | 181.699k (2.07%)   | 45.38k (0.52%)     |
> | 80-90   | 97.618k (1.11%)    | 23.823k (0.27%)    |
> | 90-100  | 53.981k (0.62%)    | 13.299k (0.15%)    |
> | 100-110 | 83.066k (0.95%)    | 21.008k (0.24%)    |
>
>
>
>
>
> We are getting the performance distribution for graphs of varying sizes (i.e., number of triples) in order to gauge whether the model generalizes well for long inputs. We will try our best to obtain and report the results within the discussion phase. Unfortunately, we couldn’t immediately fetch the results. The server room in our university that housed the GPU server used for the experiments in our submission had a recent incident which damaged the server and other servers housed there. Although we backed up the trained model, the individual sentences generated for test cases were not backed up in time. While damage assessment and recovery effort are slowly undergoing, it is unclear whether and when we could recover the data on the server. And thus we need to rerun the model on the test set on a new server. We have started doing that on a cloud computation instance.
>
>
>
> [Colas et al., 2021] "EventNarrative: A Large-scale Event-centric Dataset for Knowledge Graph-	to-Text Generation." Thirty-fifth Conference on Neural Information Processing Systems Datasets and 	Benchmarks Track (Round 1). 2021

---

### Official Review · Reviewer_RttK · 2023-08-04

**Soundness:** 4

**Excitement:**

4: Strong: This paper deepens the understanding of some phenomenon or lowers the barriers to an existing research direction.

**Paper Topic And Main Contributions:**

The academic paper introduced a new graph-to-text dataset named “GraphNarrative”, which offers a significant improvement over existing datasets by incorporating more realistic large-scale and open-domain settings. The paper further delves into a notable issue that is found within the approaches which applied pretrained language models to solve graph-to-text task – the phenomenon of information hallucination. Remarkably, this study is the first of its kind to quantitatively measure the prevalence of hallucination induced by graph-to-text models. To address this prevalent issue, the authors propose a  approach, which involves trimming the sentence according to its dependency parse tree and eliminates portions that are not represented in the graph in order to effectively eliminate hallucination. The paper reports thorough experiments and evaluations to demonstrate the utility and effectiveness of the newly introduced GraphNarrative and the novel approach proposed for combating hallucination.


**Questions For The Authors:**

Regarding the proposed hallucination mitigation method, could a sentence after trimming be semantically incomplete (due to the lack of the trimmed sub-sentences)?

Is there any possibility that the proposed sentence trimming method could falsely remove sub-sentence that aligns with the given graph (False Negatives)?



**Reasons To Accept:**

The paper stands as a pioneering research on evaluating and mitigating the information hallucination issue inherent in graph-to-text models with PLMs. Besides, the author pointed out the limitations of existing datasets, i.e., they are small hand-crafted and contain limited entity types and relations. Besides, the text descriptions in the instances of hand-crafted datasets tend to follow monotonous templates. Also, the graph fragments in existing datasets are largely limited to simple star graphs or acyclic graphs. The author argues that models trained on such data may not be as effective on large-scale and open-domain knowledge graphs in the real world. In comparison, the proposed GraphNarrative dataset contains more than 8 million graph-sentence pairs and use Wikipedia as text corpus, which has diverse narrations of various relations. GraphNarrative also contains a total of 7,920 distinct topological shapes based on graph isomorphism.

Moreover, the logical framework of the entire paper is clearly structured, the theoretical concept and experimental detail are thorough. The writing of this paper is good and easy to follow. Besides, the selected datasets are representative and the experiment settings are reasonable.


**Reasons To Reject:**

The proposed approach requires a mapping between the graph entities and Wikipedia entities. If the mapping does not exist, the approach cannot be applied.

**Reproducibility:**

4: Could mostly reproduce the results, but there may be some variation because of sample variance or minor variations in their interpretation of the protocol or method.

**Reviewer Confidence:**

4: Quite sure. I tried to check the important points carefully. It's unlikely, though conceivable, that I missed something that should affect my ratings.

**Typos Grammar Style And Presentation Improvements:**

line 031-033: does the sentence mean "The graph-to-text generation task entails the generation of a token sequence Y=(y1, ..., yn) given a subgraph to describe G"?

---

> ### Author Rebuttal · Authors · 2023-08-29
>
> We are grateful to the reviewer for the thoughtful review which will help us improve the final version of the paper.
>
>
>
> **1.** “The proposed approach requires a mapping between the graph entities and Wikipedia entities. If the mapping does not exist, the approach cannot be applied.”
>
>
>
> **Response**
>
> This is a valid concern. When such a mapping is missing, techniques for NER, information extraction, and so on can be potentially applied to create such a mapping. The impact of such an algorithmically discovered mapping on the efficacy of graph-to-text models remains to be examined.
>
>
>
> **2.** “Regarding the proposed hallucination mitigation method, could a sentence after trimming be semantically incomplete (due to the lack of the trimmed sub-sentences)?
>
>
>
> “Is there any possibility that the proposed sentence trimming method could falsely remove sub-sentence that aligns with the given graph (False Negatives)?”
>
>
>
> **Response**
>
> If our interpretation of these questions is incorrect, please let us know and we’d love to engage in further discussions. Based on our interpretation, the two questions are about the possibility of sentence trimming leading to the removal of sentence subsequences that are actually depicted in the graph. In theory there could be two corresponding scenarios:
>
>
>
> a) Entities in the graph being trimmed from the sentence: this scenario is implausible. The trimming method consistently retains entities from the graph in the sentence, as they always reside on their shortest dependency path.
>
>
>
> b) Relationships within the graph being trimmed from the sentence: This scenario is possible, although the loss is likely outweighed by the gain. Our human evaluation results (Table 3 in the paper) show a marginal increase in missed relationships, from 0.04 to 0.083, but a much more significant reduction of fabricated relationships from 1.340 to 0.453. (There was a typo in the paper---in Table 3, 0.04 and 0.083 should be swapped. We apologize for this and we will fix it in the final version.)
>
> **3.** "line 031-033: does the sentence mean "The graph-to-text generation task entails the generation of a token sequence Y=(y1, ..., yn) given a subgraph to describe G"?"
>
>
> **Response**
>
> Yes. Thank you! We will fix the grammar error.

---

### Official Review · Reviewer_TcQj · 2023-08-11

**Typos Grammar Style And Presentation Improvements:** None at the moment
**Soundness:** 3

**Excitement:**

3: Ambivalent: It has merits (e.g., it reports state-of-the-art results, the idea is nice), but there are key weaknesses (e.g., it describes incremental work), and it can significantly benefit from another round of revision. However, I won't object to accepting it if my co-reviewers champion it.

**Justification For Ethical Concerns:**

None at the moment

**Missing References:**

None at the moment

**Paper Topic And Main Contributions:**

This paper studies the problem of graph to text generation, which is widely used in search and knowledge graph query. In this paper, the author propose a new graph to text dataset which aligns with real world knowledge graphs with more complex substructurs and relation types than previous methods. Besides, to mitigation hallucination when generating, the author propose sentence triming to eliminate irrelevant contents in the generated sentence. Experimental results show the effectiveness of the proposed sentence triming methods and the quality of the datasets.

**Questions For The Authors:**

1. Can you provide some evaluation result with large scale models like GPT-3.5. I wonder if the hallucination could be ealisy mitigated with models with large scale parameters, and the proposed sentence triming method is only suitable for small models as used in this paper.

2. Will the transformer structures (encoder-only, encoder-decoder, decoder-only) affects the result ?

3. Can you provide some explanations that why sentence triming seems not so effective on WebNLG 2017 (Table 7) ?

**Reasons To Accept:**

1. Propose a ground new large scale dataset that solves the problems raised by previous datasets. The dataset contains various substructures and a large amount of relations which is aligned with real world knowledge graphs.

2. To eliminate the hallucination problem, the author propose a novel method, sentence triming, to eliminate portions that are not present in the corresponding graph, by utilizing the sentence’s dependency parse tree, which is interesting.

3. Extensive experiments are conducted through machine and human evaluations and the result almost aligned with the claimed contributions.

**Reasons To Reject:**

1. Only T-5-X and BART is evaluated on the new dataset, while the result for most recent large language models are not presented. Besides, the evaluation only conduct on LM with encoder or encoder-decoder based transformer structures, no result for decoder-only transformers models (e.g. GPT-X) is presented.

2. Experimental settings for previous dataset is not clear. The author may make a more detailed introduction to the settings of the experiments on previous datasets like TEKGEN and WebNLG 2017. For example, how the dataset is splited and is the experimental setting remains the same with baselines.

3. Sentence triming, which is the main method proposed by the author, seems not generalize well to all datasets. It shows a large performance gain on TEKGEN in Table  8. However, the performance gain seems pretty incremental in WebNLG as in Table 7.

**Reproducibility:**

4: Could mostly reproduce the results, but there may be some variation because of sample variance or minor variations in their interpretation of the protocol or method.

**Reviewer Confidence:**

4: Quite sure. I tried to check the important points carefully. It's unlikely, though conceivable, that I missed something that should affect my ratings.

---

> ### Author Rebuttal · Authors · 2023-08-29
>
> We would like to thank the reviewer for the thoughtful comments which will help us improve the camera-ready version of the paper. Please see below for our response to the comments.
>
>
>
> **1.** “Only T-5-X and BART is evaluated on the new dataset, while the result for most recent large language models are not presented. Besides, the evaluation only conduct on LM with encoder or encoder-decoder based transformer structures, no result for decoder-only transformers models (e.g. GPT-X) is presented.”
>
>
>
> and
>
>
>
> “Can you provide some evaluation result with large scale models like GPT-3.5. I wonder if the hallucination could be ealisy mitigated with models with large scale parameters, and the proposed sentence triming method is only suitable for small models as used in this paper.”
>
>
>
> **Response**
>
>
>
> Thank you for the comment. We identified a couple of studies that use GPT-X for graph-to-text tasks. (Harkous et al., 2020) used GPT-2, and a July 2023 pre-print (Yuan and Färber, 2023) used GPT-3 and chatGPT (GPT-3.5) on WebNLG and AGENDA datasets. Results from encoder-decoder models (T5, BART) still outperform the results of the decoder-only models reported in these publications. Furthermore, (Yuan and Färber, 2023) also shows that GPT-3 and GPT-3.5 still hallucinate facts. Nevertheless, we will discuss these publications and we would like to reproduce the results of using GPT-3.5, for inclusion in the camera-ready version of the paper.
>
>
>
> [Harkous et al., 2020] Hamza Harkous, Isabel Groves, and Amir Saffari. 2020. Have Your Text and Use It Too! End-to-End Neural Data-to-Text Generation with Semantic Fidelity. In Proceedings of the 28th International Conference on Computational Linguistics, pages 2410–2424, Barcelona, Spain.
>
>
>
> [Yuan and Färber, 2023] Shuzhou Yuan, Michael Färber. "Evaluating Generative Models for Graph-to-Text Generation." arXiv preprint arXiv:2307.14712 (2023).
>
>
>
>
>
> **2.** “Experimental settings for previous dataset is not clear. The author may make a more detailed introduction to the settings of the experiments on previous datasets like TEKGEN and WebNLG 2017. For example, how the dataset is splited and is the experimental setting remains the same with baselines.”
>
>
>
> **Response**
>
>
>
> We used the same split (train, dev, test) provided by the TEKGEN and WebNLG. For WebNLG, we also used their split between seen and unseen domains. Furthermore, the training strategy on TEKGEN and WebNLG are the same as in (Ribeiro et al., 2021) which used T5-large model. We will make these details clear in the final version. Details about experiment settings for linearization of the graphs and other hyperparameters are mentioned in Lines 543-546 and Lines 559-572 of the paper.
>
>
>
> [Ribeiro et al., 2021] Leonardo FR Ribeiro, Martin Schmitt, Hinrich Schütze, and Iryna Gurevych. 2021. Investigating pretrained language models for graph-to-text generation. In Proceedings of the 3rd Workshop on Natural Language Processing for Conversational AI, pages 211–227.
>
>
>
>
>
> **3.** “Sentence triming, which is the main method proposed by the author, seems not generalize well to all datasets. It shows a large performance gain on TEKGEN in Table 8. However, the performance gain seems pretty incremental in WebNLG as in Table 7.”
>
>
>
> and
>
>
>
>  “Can you provide some explanations that why sentence triming seems not so effective on WebNLG 2017 (Table 7) ?”
>
>
>
> **Response**
>
>
>
> Sentence trimming does not benefit human-annotated datasets (e.g., WebNLG, DART) much after fine-tuning on it. The main reason is that human-annotated datasets generally have good quality with graph and text well-aligned, i.e., the text exactly describes the content of the graph without extra information and very few hallucinations in their training data. Thus, they do not really need to be trimmed to remove parts that appear in text but not in the graph. On the contrary, in automatically collected datasets such as TEKGEN and GraphNarrative, the text more often contains more information than the graph, especially because real-world knowledge graphs are incomplete. For such kind of data, trimming will be more useful. We will provide more detailed analysis like this in the final version.
>
>
>
> We added the zero-shot experiments on WebNLG, results are shown in the below table. First fine-tuning on GraphNarrative improves the model’s generalization ability on WebNLG for zero-shot learning.
>
>
>
> | Model    | BLEU  | METEOR | ChrF++ |
> | -------- | ----- | ------ | ------ |
> | T5-large | 4.01  | 9.54   | 24.64  |
> | GN-T5    | 21.38 | 31.82  | 56.83  |
> | GNST-T5  | 27.60 | 32.27  | 56.81  |
>
>
>
>
>
> **4.** “Will the transformer structures (encoder-only, encoder-decoder, decoder-only) affects the result?”
>
>
>
> **Response**
>
>
>
> We have a hypothesis that the transformer structures may not affect the results, because sentence trimming doesn’t operate on the model. It only modifies the dataset. We’d like to experiment and verify this and report corresponding results in the final version. We are not sure which encoder-only model can be used for generation tasks. Does the reviewer have any suggestions?

---

### Meta-Review · Area_Chair_3q6p · 2023-09-19

**Recommendation:** 4

**Metareview:**

This paper works on generating the natural language description for knowledge graph triples. The reviewers find the proposed datasets to be useful, practical. The proposed method seems novel and extensively evaluated with human and automatic metrics. The overall paper seems logically structured well. The reviewers share concern about not including the recent language models and not studying the decoder only models. The author responses have mostly addressed the reviewers comments and the authors should include the new results in the draft to make the paper clear. A thorough investigation of when the proposed method generalizes to good accuracy need to be worked out.

---

### Decision · Program_Chairs · 2023-10-07

**Decision:**

Accept-Main

**Comment:**

This paper works on generating the natural language description for knowledge graph triples. The reviewers find the proposed datasets to be useful, practical. The proposed method seems novel and extensively evaluated with human and automatic metrics. The overall paper seems logically structured well. The reviewers share concern about not including the recent language models and not studying the decoder only models. The author responses have mostly addressed the reviewers comments and the authors should include the new results in the draft to make the paper clear. A thorough investigation of when the proposed method generalizes to good accuracy need to be worked out.